# Proactive COVID-19 Infection Prevention Measures in a Hyperbaric Oxygen Therapy Center

**DOI:** 10.3390/medicina56060261

**Published:** 2020-05-27

**Authors:** Jing-Jou Lo, Su-Chen Wang, Hsiu-Ying Lee, Su-Shin Lee, Hsiao-Chen Lee, Ching-Tzu Hung, Shu-Hung Huang

**Affiliations:** 1Department of General Medicine, Kaohsiung Medical University Hospital, Kaohsiung Medical University, Kaohsiung 807, Taiwan; ljingjou@gmail.com; 2Hyperbaric Oxygen Therapy Center, Kaohsiung Medical University Hospital, Kaohsiung Medical University, Kaohsiung 807, Taiwan; h870812@yahoo.com.tw (S.-C.W.); 910033@kmuh.org.tw (H.-Y.L.); sushin@kmu.edu.tw (S.-S.L.); s640125@yahoo.com.tw (H.-C.L.); 3Division of Plastic Surgery, Department of Surgery, Kaohsiung Medical University Hospital, Kaohsiung Medical University, Kaohsiung 807, Taiwan; 4Infection Control Center, Kaohsiung Medical University Hospital, Kaohsiung Medical University, Kaohsiung 807, Taiwan; 830293@kmuh.org.tw; 5Department of Surgery, School of Medicine, College of Medicine, Kaohsiung Medical University, Kaohsiung 807, Taiwan

**Keywords:** coronavirus disease 2019, SARS-CoV-2, infection control, cross infection, HBOT

## Abstract

Since the outbreak of coronavirus disease 2019 (COVID-19) in Wuhan, China in December 2019 and its subsequent global spread, Taiwan has been combatting this pandemic. COVID-19 is caused by a novel coronavirus, severe acute respiratory syndrome coronavirus 2 (SARS-CoV-2). As SARS-CoV-2 can be transmitted through droplets and aerosols, we cannot ignore the risk of transmission during hyperbaric oxygen therapy (HBOT). Our hyperbaric oxygen therapy center prioritizes preventing the spread of COVID-19 and maintaining operation for the patients during the pandemic. The aim of this article is to share the protocol that we have adopted in our hyperbaric oxygen therapy center to help prevent the spread of COVID-19.

Dear Editor,

On 11 March 2020, the World Health Organization (WHO) declared the coronavirus disease 2019 (COVID-19) a global pandemic. COVID-19 is caused by a novel coronavirus, severe acute respiratory syndrome coronavirus 2 (SARS-CoV-2) [1]. Since the outbreak of COVID-19 in Wuhan, China in December 2019 and its subsequent global spread, Taiwan has been combatting this pandemic [2]. As of 14 May 2020, a total of 4,248,389 confirmed COVID-19 cases with 294,046 deaths have been reported by the WHO; the total number of COVID-19 positive cases in Taiwan was only 440 with seven deaths (of the total population of around 23.57 million), despite the extreme close proximity between Taiwan and China. As of now, the majority of the confirmed COVID-19 cases involved traveling to SARS-CoV-2-infected regions, and no evidence of community transmission has been noted [3,4]. Symptoms of COVID-19 include fever, cough, shortness of breath, myalgia, diarrhea, loss of smell and taste; nevertheless, some patients do not manifest any symptoms or only have nonspecific symptoms [2,5,6,7]. The possible routes of transmission are not well studied, including respiratory droplets and contact with contaminated environmental surfaces [8]. As SARS-CoV-2 can be transmitted through droplets and aerosols, the risk of disease transmission during hyperbaric oxygen therapy (HBOT) must not be neglected, especially when individuals without any symptoms are not routinely tested for COVID-19 in Taiwan. The aim of this article is to share the protocol that we have adopted at our hyperbaric oxygen therapy center to help prevent the spread of COVID-19.

Our hyperbaric oxygen therapy center prioritizes preventing the spread of COVID-19 and maintaining its operations during the pandemic. Our strategies for minimizing the risk of infection are, first, limiting the number of healthcare professionals who come in contact with patients and, second, having the same team of health care professionals interacting exclusively with the same group of patients [9]. Our center is equipped with a multiplace chamber, HAUX-STARMED 2000/5,5 Hyperbaric Chamber, Germany, and the main chamber capacity is ten seated persons; the auxiliary chamber capacity is two seated persons. Air circulation of the chamber is effected from a heat exchanger with ventilator specially designed for pressure chambers. Chamber air is suctioned out of the chamber area, cooled down in the heat exchanger and blown in again into the upper part of the chamber. According to the database of the Kaohsiung Medical University Hospital (KMUH) healthcare system, our center mainly treats chronic wounds (38.42%), idiopathic sudden sensorineural hearing loss (20.53%), osteomyelitis (13.35%) and open wounds (11.53%) in ordinary circumstances, and after the start of the pandemic we have seen a decrease in treatment of non-critical patients.

Guidelines by Taiwan Centers for Disease Control (CDC) and the KMUH healthcare system, stipulate three epidemic levels, each with a respective epidemic prevention plan (Table 1) [10]. Epidemic level 0 refers to a first confirmed locally infected case in Taiwan that could not be traced to any clear source of infection, or a first case in Taiwan, except Kaohsiung, confirmed with nosocomial infection. During epidemic level 1, a first infected medical personnel confirmed in Kaohsiung, or a first case confirmed with nosocomial infection in Kaohsiung is reported. During epidemic level 2, a first infected medical personnel confirmed within the KMUH healthcare system, or a third infected medical personnel confirmed in Kaohsiung is reported.

The current epidemic level of the KMUH healthcare system is epidemic level 0. As we divide medical staffs into two work groups, group 1 consists of dedicated chamber operators, who must not come in contact with patients and their companions, whereas group 2 consists of dedicated nurses, who are responsible for performing at-gate body temperature measurement and infectious risk assessment including the travel, occupation, contact, and cluster (TOCC) history, checking patients’ vital signs before and after therapy and assisting patients in completing the therapy. Group members work and dine in different designated areas. In the hyperbaric oxygen chamber, patients also sit separately and stay at least one meter apart from each other, which entails a reduction in the maximum number of patients per dive from ten to five patients (Figure 1A). The total time of dive is 90 min, including 15-min descent with air, 60-min dive at an increased oxygen partial pressure of 1.5 bar or 2.5 atmosphere absolute (ATA) with 100% O_2_, and 15-min ascent with 100% O_2_. During descent, the pressure in the chamber is increased 1.5 bar by compressing gas into the chamber within 15 min, descent rate 0.1 bar/min. During ascent, compressed gas is released from the chamber within 15 min, ascent rate 0.1 bar/min. For the more severe epidemic level 1, we further divide our medical staff into two care teams. One team works in the morning, while the other works in the afternoon; each team always cares for the same set of patients, and each doctor will always be on the same team. Each team also comprises two work groups of dedicated chamber operators and dedicated nurses. The maximum number of patients per dive is also reduced to three patients (Figure 1B). For the most severe epidemic level, level 2, the duration of dive is reduced from 90 to 60 min, including 15-minute descent with 100% O_2_, 30-min dive at an increased oxygen partial pressure of 1.5 bar or 2.5 ATA with 100% O_2_, and 15-min ascent with 100% O_2_; the number of patients per dive is limited to only one patient (Figure 1C). During this period, the multiplace chamber is used as a mono chamber, and the patient breathes 100% O_2_ throughout the therapy, so the duration at an increased oxygen partial pressure of 1.5 bar or 2.5 ATA is shorter from 60 min to 30 min to avoid oxygen toxicity. When we reduce capacity per dive, we run more dives form three sections per day to up to four sections per day to meet the clinical demand in our center. Currently, the daily number of patients receiving HBOT in our center are around 12 persons, so in the epidemic level 0 and 1, the epidemic prevention plans are enough for currently clinical demand in our center. If the max daily capacity was met, HBOT for patients with Undersea & Hyperbaric Medical Society (UHMS) indications for HBOT is persisted, and these indications include air or gas embolism, carbon monoxide poisoning, cyanide poisoning, clostridial myositis and myonecrosis (gas gangrene), crush injury, compartment syndromes and other acute traumatic ischemia’s, decompression sickness, arterial insufficiencies (enhancement of healing in selected problem wound and central retinal artery occlusion), severe anemia, intracranial abscess, necrotizing soft tissue infections, osteomyelitis (refractory), delayed radiation injury (soft tissue and bony necrosis), compromised grafts and flaps, acute thermal burn injury, idiopathic sudden sensorineural hearing loss [11]; HBOT for patients with headache, cerebrovascular accident, neuritis, operation wounds after cosmetic surgery, is suspended until after the COVID-19 pandemic.

We have implemented and reinforced a set of strict regulations to protect the safety and health of both the healthcare providers and the patients. Everyone must wear a medical mask at all times, and they must wash their hands with an alcohol-based (75%) detergent before entering and leaving the center. Before therapy, nurses measure the body temperature and take TOCC history of both patients and their company at gate. The number of companions is limited to only one person per patient. For inpatients, their medical records are transferred into the center. Doctors will also assess patients, especially for upper respiratory symptoms such as cough, diarrhea of unknown reason, as well as loss of smell and taste, and examine the chest X-ray film of every new patient. During therapy, patients are not allowed to remove the oxygen mask during the entire therapy, and the duration of breathing media delivery is controlled by dedicated chamber operators so as to avoid oxygen toxicity. Each patient has a designated seat number in the hyperbaric oxygen chamber for all therapy appointments. After therapy, cleaners clean the chamber in accordance with disinfection guidelines. The surfaces that contact with patients are deep cleaned with surface disinfectants, Surfa’Safe Premium, Laboratoires Anios, France. The oxygen mask is dedicated to a single patient, and corrugated tubing is disposable. Patient contact is eliminated: even the time at and space in which patients change their clothes are purposefully separated.

When patients without TOCC history turn symptomatic in the course of HBOT, they are allowed to continue receiving HBOT; when patient with TOCC history turn symptomatic, they are not allowed in the facility and are referred to emergency room for COVID-19 testing. Positive TOCC history includes travel to SARS-CoV-2-infected regions in the previous 14 days; special occupations, e.g., navy, healthcare workers, traveling business workers, and public transportation workers; or contact with and cluster exposure to COVID-19-positive patients in the previous 14 days. If a patient diagnosed with COVID-19 receives HBOT, the other patients, their companions and healthcare personnel in the same treatments are regarded as contacts, tested for COVID-19, and quarantined for 14 days.

According to the recommendations regarding personal protective equipment in the context of COVID-19 disease from WHO, in our center, all healthcare workers wear a surgical mask, and cleaners are additionally equipped with heavy duty gloves, gown, goggles, and closed work shoes [12]. All staffs are also subject to infection prevention measures: their body temperature is measured twice daily, and they are closely monitored for signs of COVID-19, e.g., fever, cough, diarrhea of unknown reason, as well as loss of smell and taste. They must also wash their hands frequently with alcohol-based (75%) detergent, frequently change their work attire, and work and dine in separate areas.

Proactive infection prevention during the COVID-19 pandemic can preserve medical personnel and medical resources to ensure the ability to provide sufficient health care to the entire society. In sharing our experience, we hope to help the rest of the world in their fight against COVID-19.

## Figures and Tables

**Figure 1 medicina-56-00261-f001:**
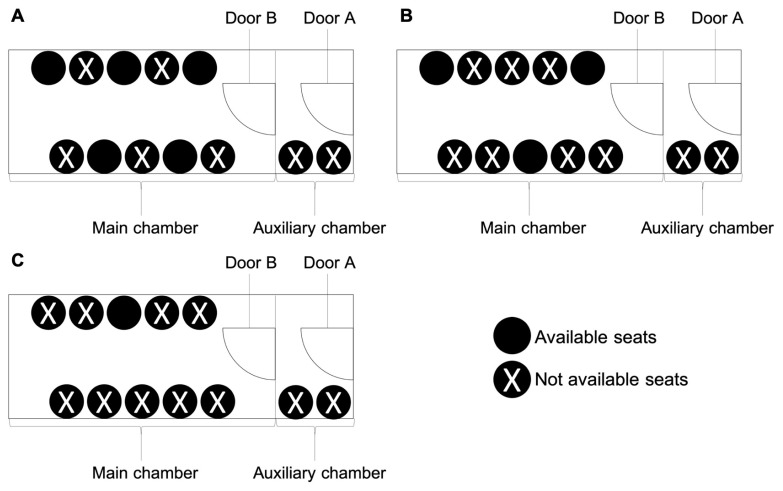
Seating plan in the hyperbaric oxygen chamber for epidemic level 0 (**A**), epidemic level 1 (**B**), and epidemic level 2 (**C**).

**Table 1 medicina-56-00261-t001:** Plans for different epidemic levels.

Epidemic level	0	1	2
**Definition**	First confirmed locally infected case in Taiwan that could not be traced to any clear source of infection, or first case in Taiwan, except Kaohsiung, confirmed with nosocomial infection	First infected medical personnel confirmed in Kaohsiung, or first case confirmed with nosocomial infection in Kaohsiung	First infected medical personnel confirmed within the KMUH healthcare system, or the third infected medical personnel confirmed in Kaohsiung
**Work group**	Yes	Yes	Yes
**Fixed caring team**	No	Yes	Yes
**Patients per dive**	5	3	1 ^‡^
**Total time of dive**	90 min	90 min	60 min
**Duration of descent/breathing media ***	15 min/air	15 min/air	15 min/100% O_2_
**Duration at an increased oxygen partial pressure of 1.5 bar or 2.5 ATA/breathing media**	60 min/100% O_2_	60 min/100% O_2_	30 min/100% O_2_
**Duration of ascent/breathing media ^†^**	15 min/100% O_2_	15 min/100% O_2_	15 min/100% O_2_

ATA: atmosphere absolute; KMUH: Kaohsiung Medical University Hospital; O_2_: oxygen. * During descent, the pressure in the chamber is increased 1.5 bar by compressing gas into the chamber within 15 min, descent rate 0.1 bar/min. ^†^ During ascent, compressed gas is released from the chamber within 15 min, ascent rate 0.1 bar/min. ^‡^ During epidemic level 2, the multiplace chamber is used as a mono chamber, and the patient breathes 100% O_2_ throughout the therapy, so the duration at an increased oxygen partial pressure of 1.5 bar or 2.5 ATA is shorter from 60 min to 30 min to avoid oxygen toxicity.

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
