# Peer review of "Proactive COVID-19 Infection Prevention Measures in a Hyperbaric Oxygen Therapy Center"

_medicina, 2020, doi:10.3390/medicina56060261_

Round 1
Reviewer 1 Report
Dear colleagues,
Thank you for your interesting letter. You provide your local protocol to continue HBOT during the COVID-19 pandemic. Which is good to share with colleagues, to broaden (often only national) perspectives.
However, I have quite a few remarks and suggestions to the text. Attached you can find a PDF with comments, but I will mention a few important ones here. Again, more information is added in the PDF
- Seven authors for this letter seems inappropriate for a letter to the editor that consists of about 50 lines. Please adhere to ICMJE guidelines for authorship. From face value I would suggest 2 or 3 authors, but I leave that up to the editor.
- According to the Coronaviridae Study Group COVID-19 should be referenced to as SARS-CoV-2. Link is added in the PDF.
- The numbers mentioned in line 32-33 are already one month old. Please update them to the most recent values you can find (perhaps May 1 2020)?
- What is the current epidemic level in Taiwan? This gives the reader context for your perspective on HBOT.
- Do you really compress patients to 50 msw/165 fsw with 100% oxygen? Most HBOT regimes to up to 15 msw due to oxygen toxicity. Please confirm this.
- When you reduce capacity per treatment session, do you run more sessions or do you treat less patients? I expect the latter. Could you elaborate on which patients continue to receive HBOT, and which patients are denied HBOT until after the COVID-19 pandemic?
- Another remark, which probably seems obvious, but is missing from the letter: what do you do when patients turn symptomatic in the course of HBOT treatments? Do you consider the patients in the same treatments as infected? And what about healthcare personnel that came into contact with them? Is someone with cough (but without fever) allowed in the facility? Please share your thoughts.
- References 2, 4 and 5 are incorrectly abbreviated (probably by the reference manager). This should be corrected.
Again, thank you for offering this letter. As sharing of insights is important to quickly adapt (worldwide) to the 'current reality' of COVID-19. However, I feel this letter requires substantial changes before it should be published.
(unblinded, as I feel all reviews of scientific papers should be)

Reviewer 2 Report
Dear editor,
Thank you for letting me review this paper, describing hyperbaric oxy therapy during the COVID-19 pandemic.
The subject is of high interest, I've only few minor corrections in the attached pdf, good job!

Round 2
Reviewer 1 Report
Dear colleagues,
Thank you for thoroughly revising the manuscript and addressing the comments.
I have just a few (closing) remarks, please also see the attachement:
- I feel lines 62 to 78 could be heavily compressed to just the main message you want to tell: our center mainly treats X, Y and Z in ordinary circumstances and after the start of the pandemic we have seen a decrease in treatment of non-critical patients. It currently holds, in my opinion, too much data and percentages which do not add to the message you want to deliver with your paper.
- The footnote markings in table 1 should be '*, †, ‡, §, |, ¶', to avoid confusion with references to literature.
- Please review your reference list, there are a few details that should be corrected.
